# Self-reported dyspareunia and outcome satisfaction after spontaneous second-degree tear compared to episiotomy: A register-based cohort study

Mette L. Josefsson[1]*, Sara Sohlberg[1], Cecilia Ekéus[1], Eva Uustal[2], Maria Jonsson[1]

**1** Department of Women's and Children's Health, Uppsala University, Uppsala, Sweden, **2** Department of Biomedical and Clinical Sciences, Linköping University, Linköping, Sweden

* mette.josefsson@uu.se

## Abstract

### Introduction

Symptoms after second-degree tears and in particular episiotomies are common. Our aim was to investigate the prevalence and degree of dyspareunia and level of satisfaction with the outcome of the perineal repair after a spontaneous second-degree tear compared to an episiotomy. Further, we aimed to identify risk factors for dyspareunia and dissatisfaction with the outcome.

### Material and methods

This register-based cohort study included 5 328 primiparous women who sustained a spontaneous second-degree tear (n = 4 323) or an episiotomy (n = 1005) between 2014 and 2019 in Sweden. The primary outcomes were self-reported degree of dyspareunia and level of satisfaction with the outcome of the perineal repair at one year. Data were collected from national health and quality registers and online questionnaires at eight weeks and one year. Logistic regression was used and results are presented by Odds Ratios (OR) with 95% confidence intervals (CI) after adjustment for age, body mass index and mode of delivery.

### Results

30.0% of women with a spontaneous tear and 29.1% of women with an episiotomy reported mild or moderate dyspareunia, while 2.4% of women with a spontaneous tear compared to 3.8% of women with an episiotomy reported strong or unbearable dyspareunia (aOR 1.5; CI 0.9–2.4). 73.4% of women with a spontaneous tear and 67.1% with episiotomy were satisfied or very satisfied with their outcome, while 6.7% with an episiotomy compared to 3.7% with a spontaneous tear were dissatisfied (aOR 1.8; CI 1.2–2.6). Postpartum infection, scar dehiscence, re-suturing and perineal pain at eight weeks were risk factors for dyspareunia and dissatisfaction at one year.

**Data Availability Statement:** The register data cannot be shared publicly because of Swedish privacy laws. The data for this research project has

been exported from the National Board of Health and Welfare in Sweden, which does not permit data-sharing according to the Swedish Secrecy Act. Data is available from the Swedish National Board of Health and Welfare (contact per email via registerservice@socialstyrelsen.se) for researchers who meet the criteria for access to confidential data.

**Funding:** MLJ received funding by the Region Uppsala Research and Development Grant and by the General Maternity Hospital Foundation. The sponsors did not play any role in the study design, data collection and analysis, decision to publish, and preparation of the manuscript.

**Competing interests:** The authors have declared that no competing interest exist.

## Conclusions

Approximately one-third of women with either a spontaneous tear or an episiotomy reported mild or moderate dyspareunia at one year, while strong or unbearable pain was uncommon in both groups. The majority of women were satisfied or very satisfied with the outcome although episiotomy more often predicted dissatisfaction.

## Introduction

Perineal trauma during a woman's first vaginal delivery is common. Approximately 60–85% of women sustain a spontaneous second-degree perineal tear including episiotomies [1–4]. Perineal tears are divided into first-degree which includes the vaginal mucosa or skin, second-degree which also involves the perineal muscles, and third- and fourth-degree which include in addition the anal sphincter [1].

Sexual function is important for quality of life and complaints such as dyspareunia and impaired sexual satisfaction are common among women during the first year postpartum [5–9]. Instrumental delivery, perineal tears, and episiotomies have been suggested as negatively impacting sexual function, although studies are small and results sometimes conflicting [7, 10]. Several studies have focused on length of time until resumed sexual intercourse and report that perineal trauma is associated with a delayed resumption of vaginal intercourse at six months after childbirth [11].

Dyspareunia is a complaint of persistent or recurrent pain or discomfort associated with attempted or completed vaginal penetration [12]. Postpartum dyspareunia is common, with frequencies varying between 20–40% at six months [5–9] and women with a second-degree tear are 80% more likely to report dyspareunia compared to women with an intact perineum [6]. The Perineal Study in Norway that included 561 respondents did not find that episiotomy compared to spontaneous second-degree tear was a risk factor for dyspareunia, however the numbers were small; only 8% had a spontaneous second-degree tear and 21% an episiotomy, while the rest had either intact perineum, first- or third-degree tears [13]. In contrast, the recent meta-analysis by Cattani et al. that included studies with sample size between 108 and 832 women, and a range of follow-up between five and 15 months, found that episiotomy compared with spontaneous tear was associated with dyspareunia [10].

Given that previous studies have conflicting results regarding the rate of dyspareunia after second degree tears and episiotomies, and that the degree of dyspareunia is not yet thoroughly described in the medium term, we performed a register-based study.

Roman et al. highlighted in their recent qualitative meta-synthesis the importance of including the woman's experience when conducting research on perineal tears [14]. We therefore included women's self-reported level of satisfaction with the outcome of the perineal tear as part of this research.

This study primarily aimed to investigate the prevalence and the degree of dyspareunia and to examine women's self-reported level of satisfaction with the outcome of the perineal repair one year after vaginal delivery among primiparous women. Our secondary aim was to identify risk factors for dyspareunia and for worse satisfaction with the outcome at one year postpartum.

## Material and methods

This was a Swedish register-based cohort study consisting of a total of 5 328 primiparous women who sustained a spontaneous second-degree tear or an episiotomy between January 1st

2014 and December 31$^{st}$ 2019. Ethical approval was obtained prior to starting the study (reference number 2020–03763 and 2020–06877). The 'Strengthening the reporting of observational studies in epidemiology' (STROBE) checklist was used when planning and conducting the study. Data were retrieved from three Swedish national health and quality registers; the Perineal Laceration Register (PLR), the Medical Birth Register (MBR) and the National Patient Register (NPR). All register data were preceded by consent from the women. The dataset was obtained on June 13$^{th}$ 2021. All data were anonymized and authors did not have access to information that could identify individual participants during or after data collection.

The PLR is a part of the National Quality Register of Gynecological Surgery which is a validated quality register that includes all major gynecological surgery performed in Sweden [15, 16]. The PLR was established in 2014 and was initially developed to include women with third- and fourth-degree tears, however many clinics also report some or all of their second-degree tears and episiotomies [17]. In our study, data were collected from 26 out of a total of 43 delivery units in Sweden. Almost half of the data came from three hospitals (one university hospital and two district general hospitals) that report >95% of their spontaneous second-degree tears and episiotomies. Birth and neonatal data were entered from charts manually by a nurse or a medical secretary directly into the register. Self-reported data were collected via questionnaires that were sent electronically to all included participants at eight weeks and one year postpartum. The overall response rate was 77.2% at eight weeks and 69.1% at one year, although exact response rate varies with each question. The questionnaire comprises a wide variety of questions, including self-reported level of satisfaction after perineal repair, level of sexual activity and function including degree of dyspareunia, urinary- and bowel symptoms and complications. Patient focus groups testing the PLR emphasized the need for brevity to obtain completed questionnaires. Key questions from validated forms about pelvic floor dysfunction have been selected and validated for content, context and reliability [18].

The nation-wide Swedish Medical Birth Register (SMBR) was established in 1973 and includes data on five million pregnancies and births in Sweden. Maternal data are collected at routine antenatal visits and include self-reported information on obstetric history, general health and height, as well as measured weight. Birth and neonatal data are extracted from standardized regional electronic health records and include outcomes such as gestational age, birthweight, mode of delivery, intrapartum pain relief and type of perineal tear. The SMBR includes high validity data due to semi-automated data extraction, mandatory reporting to the register and universal free access to maternity health care in Sweden [19]. As a result, 98% of all women who give birth in Sweden are included in the register. The unique personal identification numbers of mothers and offspring enable linkage of data between registers.

The NPR is a mandatory national health register that includes data on all inpatient admissions since 1987 and all outpatient secondary care visits since 2001, from both private and public healthcare specialist institutions in Sweden. The register is validated with 98% of medical records achieving correct coding at cross-check with hospital notes [20]. The register includes data such as date of admission, length of hospital stay, main diagnosis and procedures. It does not include data on primary care visits.

Women who gave birth vaginally between 2014 and 2019 were identified via the SMBR (n = 580 748) and data were linked to the PLR and NPR. Women with previous pregnancies, multiple pregnancies, women with no perineal tear or other tears than second-degree or episiotomies and those not included in the PLR were excluded. Furthermore, all women with a diagnosis of third- or fourth-degree tears, or suturing involving the internal or external sphincter, were sought via the three registers and excluded once identified to minimize misclassification. The final study population included primiparous women with a second-degree perineal tear or episiotomy (Fig 1).

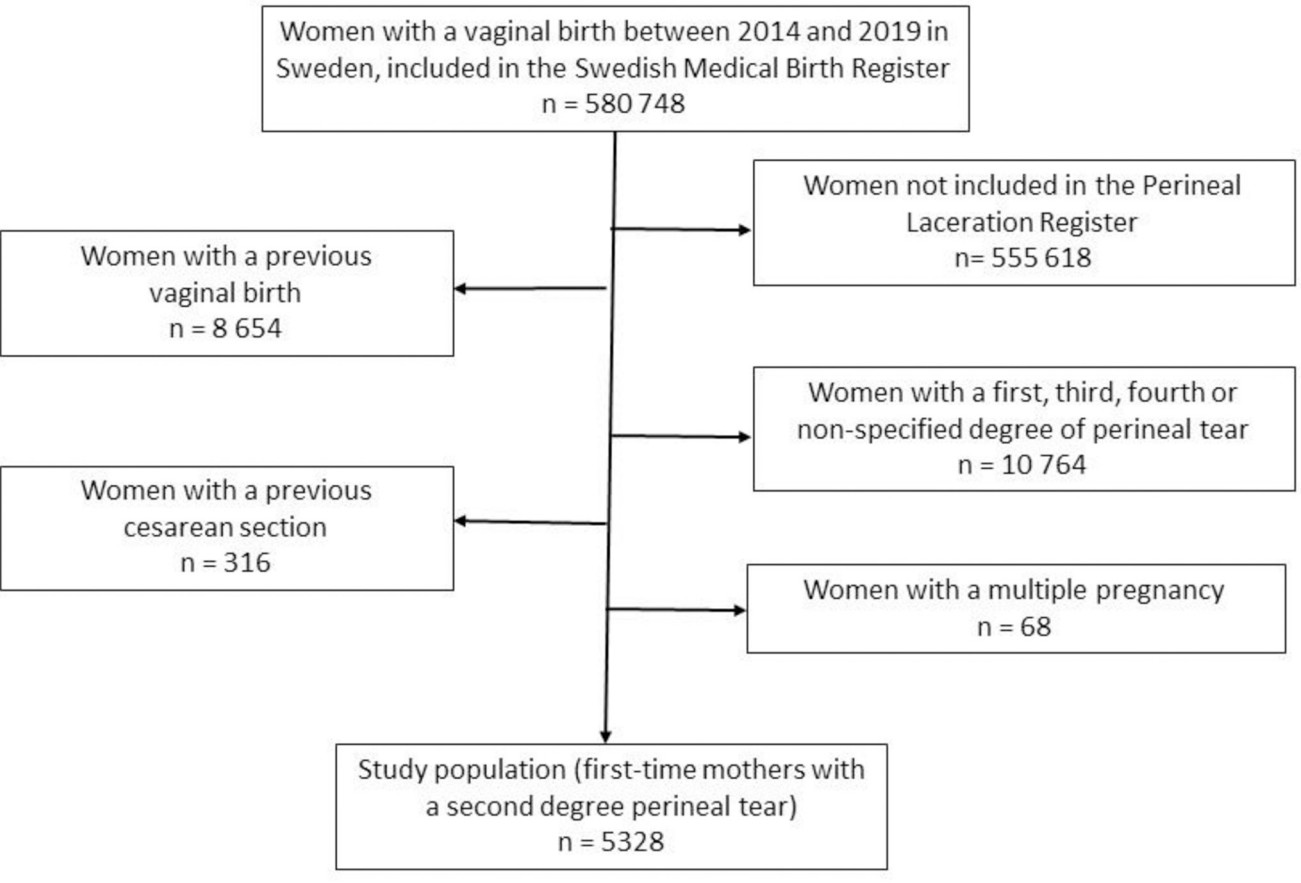

**Fig 1. Flow chart of study inclusion.**

Maternal and labor characteristics were identified via the SMBR because of its high validity data and included; age, co-morbidities, fetal presentation, mode of delivery and birth weight. The variables on co-morbidities include diagnoses prior to pregnancy such as hypertension and diabetes mellitus.

The PLR was used for the variables health care professionals performing the primary suturing, Body Mass Index (BMI) and length of active second stage. BMI (kg/m$^2$) was categorized into; <18.5 (underweight), 18.5–24.9 (normal weight) and ≥25.0 (over weight), according to the classification by the World Health Organization [21]. As is often the case with registers all data did not correlate to 100% between the three registers. We choose to define all women with an episiotomy from either the PLR (958 women), the SMBR (additional 41 women) or the NPR (additional 6 women) as having had an episiotomy (total n = 1005). To be registered as episiotomy the person filling in the data must actively choose episiotomy, hence it is likely to be correct. Women with both a spontaneous tear and an episiotomy were classified as 'episiotomy', as it is not possible from the registers to identify women with only episiotomy versus women with an episiotomy and a spontaneous tear.

Data on labor complications were all obtained from the NPR through the international classification of disease, 10$^{th}$ version (ICD-10). Postpartum complications (postpartum infection, scar dehiscence and infected scar) were also identified via the NPR, apart from 'self-reported urinary tract infection' and 'self-reported re-suturing' which were extracted from the eight-

week questionnaire sent to participants via the PLR. Frequency of self-reported data on re-suturing was higher than the data on re-suturing identified via the NPR thus we chose the data from the PLR as participants are likely to correctly remember relatively recent re-suturing.

The outcomes 'perineal pain at eight weeks', 'dyspareunia' and 'satisfaction with the outcome' were obtained from the questionnaires sent via the PLR. The Assessment of dyspareunia was included in the questionnaire at one year postpartum. Participants were asked if they experienced pain during vaginal intercourse and were given the following replies to choose from; 'no pain', 'mild pain', 'moderate pain', 'severe pain', 'unbearable pain', or 'not relevant' (not engaging in vaginal intercourse). Answers were re-categorized into three subgroups prior to statistical analysis; 'no pain', 'mild/moderate pain', 'severe/unbearable pain', while missing or 'not relevant' were excluded from the analysis. Regarding the women's satisfaction with the result of the perineal repair, participants were asked to rate their outcome by choosing one of the following options; 'very satisfied, 'satisfied', 'neutral', 'dissatisfied', or 'very satisfied'. Answers were re-categorized into three subgroups in the risk estimate analysis; 'very satisfied/satisfied', 'neutral' and 'dissatisfied/very dissatisfied'. Response rate for the question on dyspareunia was 67.8% among women with spontaneous tear and 75.8% among women with episiotomy. Response rate on satisfaction with the outcome was 65.1% for women with spontaneous tear and 71.4% for women with episiotomy.

Data were analysed using the Statistical Package for Social Sciences (SPSS 28; IBM, Armonk, NY, USA), R package (version 4.3.2) and the Modern Applied Statistics with S package (version 7.3) Numerical variables were analysed by Student t-test if normal distribution and the Mann-Whitney U-test if non-normal distribution. Categorical variables were analysed by the Pearson $X^2$ test. A value of $p<0.05$ was considered significant. All data are reported as a number, percentage, median and standard deviation (SD). Logistic regression was used to analyse association between main outcomes and spontaneous tear/episiotomy, and presented by Odds Ratio (OR) with 95% confidence interval (CI), with spontaneous tear as reference. Logistic regression was used to analyse the dependence between the 'worst outcome' ('strong/unbearable dyspareunia' and 'dissatisfied/very dissatisfied'), a binary outcome (yes or no), with different exposures (mode of delivery, perineal pain at eight weeks, postpartum infection, scar dehiscence, re-operation and BMI) and how these differed between spontaneous tear and episiotomy. Results were adjusted for mode of delivery, age and BMI, and presented as crude and adjusted OR with 95% CI. It was considered to be sufficient to adjust for mode of delivery only and not length of second stage which is a similar confounding factor.

## Results

In our cohort of 5 328 primiparous women, 81.1% (n = 4 323) sustained a spontaneous second-degree tear, while 18.9% (n = 1005) had an episiotomy. Maternal and labor characteristics are displayed in Table 1. Women who had an episiotomy compared to a spontaneous tear were older, had lower BMI and were more often delivered by vacuum extraction. Labor complications were more common in the episiotomy group with significant differences for all complications. Postpartum complications such as infection, scar dehiscence and self-reported re-suturing were also more common in the episiotomy group. 63% of the spontaneous tears and 45% of the episiotomies were sutured by midwives, the rest by physicians, which reflects the routine practice that physicians suture more complicated tears.

Table 2 describes self-reported degree of dyspareunia and level of satisfaction with the result of the perineal repair at one year in women with spontaneous tears or episiotomies. A high proportion of women reported sexual intercourse in the last three months; 85.1% after spontaneous tear and 83.0% after episiotomy (p = 0.53). The majority of women reported no

**Table 1. Maternal and labor characteristics, including labor and postpartum complications.**

| | Spontaneous tear | Episiotomy | p value |
|---|---|---|---|
| | n (%) | n (%) | |
| | n = 4 323 | n = 1 005 | |
| Maternal age (years), mean (SD) | 29.2 (4.3) | 30,0 (4.6) | <0.001 |
| Early pregnancy BMI (kg/m$^2$), mean (SD) | 24.5 (10.3) | 23.2 (4.20) | <0.001 |
| BMI (kg/m$^2$) | | | <0.001 |
| <18.5 | 104 (2.4) | 44 (4.4) | |
| 18.5–24.9 | 2 442 (56.5) | 622 (61.9) | |
| ≥25.0 | 1 595 (36.9) | 289 (28.8) | |
| Missing | 182 (4.2) | 50 (5.0) | |
| Maternal co-morbidities | | | |
| Diabetes mellitus | 25 (0.6) | 9 (0.9) | 0.26 |
| Hypertension | 13 (0.3) | 1 (0.1) | 0.49 |
| Length of active second stage (min, mean ± SD) | 40.6 (36.6) | 53.1 (54.3) | <0.001 |
| Mode of delivery | | | |
| Spontaneous vaginal | 3 539 (81.9) | 582 (57.9) | <0.001 |
| Vacuum-assisted delivery | 620 (14.3) | 383 (38.1) | <0.001 |
| Forceps | 0 (0.0) | 3 (0.3) | <0.001 |
| Missing | 164 (3.8) | 37 (3.7) | |
| Fetal presentation | | | <0.001 |
| Occiput anterior | 4 036 (93.4) | 895 (89.1) | |
| Occiput posterior | 162 (3.7) | 65 (6.5) | |
| Breech/Other | 103 (2.4) | 41 (4.1) | |
| Missing | 22 (0.5) | 4 (0.4) | |
| Birth weight (g), mean (SD) | 3 518 (471.7) | 3 524 (517.0) | 0.73 |
| Labor complications | | | |
| Labor dystocia | 938 (21.7) | 448 (44.6) | <0.001 |
| Prolonged second stage | 260 (6.0) | 82 (8.2) | 0.01 |
| Threatening fetal distress | 282 (6.6) | 191 (19.1) | <0.001 |
| Postpartum complications | | | |
| Self-reported urinary tract infection | 70 (2.0) | 30 (3.5) | 0.01 |
| Postpartum infection | 144 (3.3) | 83 (8.3) | <0.001 |
| Scar dehiscence | 25 (0.6) | 53 (5.3) | <0.001 |
| Infected scar | 4 (0.1) | 14 (1.4) | <0.001 |
| Self-reported re-suturing | 64 (1.9) | 33 (3.9) | <0.001 |

dyspareunia (64.2% vs. 61.0%), while 2.4% of women with a spontaneous tear compared to 3.8% of women with an episiotomy reported strong/unbearable dyspareunia (aOR 1.5; CI 0.9–2.4). More women with an episiotomy choose the option 'not relevant' (aOR 2.0; CI 1.3–2.9).

Overall 42.4% of women with spontaneous tears and 37.2% of women with an episiotomies reported that they were very satisfied with their outcome at one year (aOR 0.9; CI 0.7–1.0), while approximately 33% in each group reported that they were satisfied. 3.7% of women were dissatisfied compared to 6.7% of women with an episiotomy (aOR 1.8; CI 1.2–2.6), while 0.9% with spontaneous tear compared to 2.4% with episiotomy answered that they were very dissatisfied (aOR 2.4; CI 1.2–4.7), however the overall numbers were small.

Table 3 describes strong or unbearable dyspareunia at one year for different exposures. Compared to women with a spontaneous vaginal delivery, women with vacuum-assisted

**Table 2. Self-reported intercourse the last three months, dyspareunia at one year and satisfaction with the outcome of the perineal repair at one year postpartum in women who sustained a spontaneous tear or an episiotomy.**

|  | Spontaneous tear | Episiotomy | OR (95% CI) | aOR[a] (95% CI) |
|---|---|---|---|---|
|  | n (%) | n (%) |  |  |
| Dyspareunia at 1 year |  |  |  |  |
| No pain | 1 888 (64.2) | 465 (61.0) | reference | reference |
| Mild/moderate | 881 (30.0) | 222 (29.1) | 1.0 (0.9–1.2) | 1.0 (0.8–1.2) |
| Strong/unbearable | 72 (2.4) | 29 (3.8) | 1.6 (1.1–2.5) | 1.5 (0.9–2.4) |
| Not relevant | 98 (3.3) | 46 (6.0) | 1.9 (1.3–2.7) | 2.0 (1.3–2.9) |
| Satisfaction with the outcome at 1 year |  |  |  |  |
| Very satisfied | 1 194 (42.4) | 267 (37.2) | 0.9 (0.7–1.1) | 0.9 (0.7–1.0) |
| Satisfied | 953 (33.9) | 243 (33.8) | reference | reference |
| Neutral | 539 (19.1) | 143 (19.9) | 1.0 (0.8–1.3) | 1.0 (0.8–1.3) |
| Dissatisfied | 105 (3.7) | 48 (6.7) | 1.8 (1.2–2.6) | 1.8 (1.2–2.6) |
| Very dissatisfied | 24 (0.9) | 17 (2.4) | 2.8 (1.5–5.3) | 2.4 (1.2–4.7) |

[a]Adjusted for age, BMI and mode of delivery.

delivery and episiotomy were more than twice as likely to report 'strong or unbearable dyspareunia' (aOR 2.42; CI 1.29–4.25). Women with perineal pain at eight weeks more often reported dyspareunia at one year after both spontaneous tear and episiotomy and were almost four times as likely to report 'strong or unbearable' dyspareunia after spontaneous tear and four times as likely after episiotomy (aOR 4.00, CI 1.95–7.77).

Women with episiotomy and postpartum infection were three times more likely to report 'strong or unbearable' pain (aOR 3.30; CI 1.10–7.99) than women with spontaneous tear and no infection. Scar dehiscence was associated with an increased odds of reporting strong/unbearable pain among both spontaneous tear and episiotomy, however only statistically significant among the episiotomy group. Re-suturing increased the odds of dyspareunia, however overall numbers were small and the confidence interval was wide and non-significant. An episiotomy compared to a spontaneous tear in women with BMI 18.5 to 24.9 was associated with strong or unbearable pain. No differences were found among other BMI classes.

Table 4 describes self-reported satisfaction with the outcome at one year for different exposures. Women with an episiotomy were more likely to be 'dissatisfied or very dissatisfied' regardless of mode of delivery (aOR for spontaneous tear 2.08, CI 1.38–3.07 and aOR for episiotomy 2.40, CI 1.50–3.71). Women with perineal pain at eight weeks were also more dissatisfied at one year and were four times as likely to be dissatisfied or very dissatisfied after spontaneous tear and seven times as likely after episiotomy (aOR 7.80, CI 4.90–12.30). Infection was associated with increased odds of being dissatisfied regardless of spontaneous tear or episiotomy. Women with scar dehiscence and spontaneous tear were five times as likely to be dissatisfied or very dissatisfied after both spontaneous tear and episiotomy. Women with scar dehiscence following a spontaneous tear had a five times higher odds of being 'dissatisfied or very dissatisfied' compared to women with a spontaneous tear and no scar dehiscence, while the odds for women with scar dehiscence following an episiotomy was eight times higher.

Re-suturing was associated with worse satisfaction with the outcome after both spontaneous tears and episiotomies. Women with re-suturing after a spontaneous tear were three times as likely to report 'dissatisfied or very dissatisfied' while women with re-suturing after an episiotomy were seven times as likely (aOR 7.16; CI 2.52–17.82). An episiotomy compared to

**Table 3. Self-reported strong/unbearable dyspareunia at one year for different exposures.**

| Exposure | | All population | Strong/ unbearable dyspareunia | | |
|---|---|---|---|---|---|
| | | n (%) | n (%[a]) | cOR (95% CI) | AOR[b] (95% CI) |
| **Mode of delivery** | | | | | |
| Spontaneous vaginal | Spontaneous tear | 2 448 (68.8) | 58 (2.4) | reference | reference |
| | Episiotomy | 435 (12.2) | 14 (3.2) | 1.37 (0.73–2.41) | 1.39 (0.74–2.47) |
| Vacuum-assisted | Spontaneous tear | 393 (11.1) | 14 (3.6) | 1.52 (0.81–2.68) | 1.53 (0.79–2.74) |
| | Episiotomy | 281 (7.9) | 15 (5.3) | 2.32 (1.25–4.05) | 2.42 (1.29–4.25) |
| **Perineal pain at eight weeks** | | | | | |
| No pain | Spontaneous tear | 2 212 (65.8) | 35 (1.6) | reference | reference |
| | Episiotomy | 487 (14.5) | 15 (3.1) | 1.98 (1.04–3.58) | 1.80 (0.93–3.36) |
| Pain | Spontaneous tear | 464 (13.8) | 27 (5.8) | 3.84 (2.28–6.40) | 3.81 (2.23–6.43) |
| | Episiotomy | 197 (5.9) | 13 (6.6) | 4.39 (2.21–8.25) | 4.00 (1.95–7.77) |
| **Postpartum infection** | | | | | |
| No infection | Spontaneous tear | 2 750 (77.3) | 68 (2.5) | reference | reference |
| | Episiotomy | 658 (18.5) | 24 (3.6) | 1.49 (0.91–2.36) | 1.35 (0.81–2.21) |
| Infection | Spontaneous tear | 91 (2.6) | 4 (4.4) | 1.81 (0.54–4.51) | 1.93 (0.58–4.83) |
| | Episiotomy | 58 (1.6) | 5 (8.6) | 3.72 (1.27–8.77) | 3.30 (1.10–7.99) |
| **Scar dehiscence** | | | | | |
| No scar dehiscence | Spontaneous tear | 2 824 (79.4) | 71 (2.5) | reference | reference |
| | Episiotomy | 673 (18.9) | 25 (3.7) | 1.50 (0.92–2.35) | 1.34 (0.80–2.16) |
| Scar dehiscence | Spontaneous tear | 17 (0.5) | 1 (5.9) | 2.42 (0.13–12.13) | 2.87 (0.16–14.54) |
| | Episiotomy | 43 (1.2) | 4 (9.3) | 3.98 (1.17–10.22) | 3.88 (1.13–10.21) |
| **Re-suturing** | | | | | |
| No re-suturing | Spontaneous tear | 2 793 (78.5) | 69 (2.3) | reference | reference |
| | Episiotomy | 692 (19.5) | 28 (4.0) | 1.66 (1.05–2.57) | 1.50 (0.92–2.40) |
| Re-suturing | Spontaneous tear | 48 (1.3) | 3 (6.3) | 2.63 (0.63–7.43) | 2.99 (0.71–8.53) |
| | Episiotomy | 24 (0.7) | 1 (4.2) | 1.72 (0.10–8.33) | 1.48 (0.08–7.28) |
| BMI (kg/m$^2$) | | | | | |
| <18.5 | Spontaneous tear | 55 (1.6) | 4 (7.3) | 3.08 (0.90–8.02) | 3.15 (0.92–8.22) |
| | Episiotomy | 30 (0.9) | 2 (6.7) | 2.81 (0.44–9.79) | 2.55 (0.40–8.96) |
| 18.5–24.9 | Spontaneous tear | 1 612 (47.2) | 40 (2.3) | reference | reference |
| | Episiotomy | 446 (13.1) | 23 (5.2) | 2.14 (1.25–3.58) | 1.9 (1.09–3.24) |
| ≥25.0 | Spontaneous tear | 1 067 (31.2) | 24 (2.3) | 0.90 (0.53–1.50) | 0.92 (0.54–1.53) |
| | Episiotomy | 207 (6.1) | 4 (1.9) | 0.77 (0.23–1.95) | 0.70 (0.21–1.78) |

[a]Percentage within each subgroup that reported strong/unbearable dyspareunia.

[b]Adjusted for age, BMI and mode of delivery for all exposure groups apart from Mode of delivery (adjusted for age and BMI) and BMI (adjusted for age and mode of delivery).

spontaneous tear in women with BMI 18.5 to 24.9 was associated with being dissatisfied or very dissatisfied. No differences were found among other BMI classes.

## Discussion

The majority of primiparous women, regardless of whether they had a spontaneous second-degree tear or an episiotomy, reported no dyspareunia and high level of satisfaction with the outcome of their perineal repair at one year after childbirth. However, among the minority who were dissatisfied, those with episiotomy were overrepresented. The frequency of strong or

**Table 4. Self-reported 'dissatisfied or very dissatisfied' with the outcome of the perineal repair at one year for different exposures.**

| Exposure | | All population | Dissatisfied or very dissatisfied | | |
|---|---|---|---|---|---|
| | | n (%) | n (%[a]) | OR (95% CI) | AOR[b] (95% CI) |
| **Mode of delivery** | | | | | |
| Spontaneous vaginal | Spontaneous tear | 2 429 (68.8) | 106 (4.4) | reference | reference |
| | Episiotomy | 435 (12.3) | 37 (8.5) | 2.04 (1.36–2.98) | 2.08 (1.38–3.07) |
| Vacuum-assisted | Spontaneous tear | 386 (11.0) | 23 (6.0) | 1.39 (0.85–2.17) | 1.41 (0.86–2.24) |
| | Episiotomy | 283 (8.0) | 28 (9.9) | 2.41 (1.53–3.67) | 2.40 (1.50–3.71) |
| **Perineal pain at eight weeks** | | | | | |
| No pain | Spontaneous tear | 2 197 (65.6) | 66 (3.0) | reference | reference |
| | Episiotomy | 483 (14.4) | 23 (4.8) | 1.61 (0.97–2.58) | 1.59 (0.93–2.62) |
| Pain | Spontaneous tear | 466 (13.9) | 51 (10.9) | 3.97 (2.70–5.79) | 4.20 (2.82–6.23) |
| | Episiotomy | 203 (6.1) | 38 (18.7) | 7.44 (4.81–11.37) | 7.80 (4.90–12.30) |
| **Postpartum infection** | | | | | |
| No infection | Spontaneous tear | 2 723 (77.1) | 118 (4.3) | reference | reference |
| | Episiotomy | 659 (18.7) | 53 (8.0) | 1.93 (1.37–2.69) | 1.85 (1.28–2.63) |
| Infection | Spontaneous tear | 92 (2.6) | 11 (12.0) | 3.00 (1.48–5.55) | 2.75 (1.30–5.23) |
| | Episiotomy | 59 (1.7) | 12 (20.3) | 5.64 (2.79–10.58) | 4.81 (2.28–9.35) |
| **Scar dehiscence** | | | | | |
| No scar dehiscence | Spontaneous tear | 2 798 (79.2) | 125 (4.5) | reference | reference |
| | Episiotomy | 677 (19.2) | 53 (7.8) | 1.82 (1.29–2.52) | 1.70 (1.18–2.41) |
| Scar dehiscence | Spontaneous tear | 17 (0.5) | 4 (23.5) | 6.58 (1.83–18.90) | 5.37 (1.21–17.00) |
| | Episiotomy | 41 (1.2) | 12 (29.3) | 8.85 (4.26–17.35) | 8.04 (3.71–16.32) |
| **Re-suturing** | | | | | |
| No re-suturing | Spontaneous tear | 2 766 (78.3) | 122 (4.4) | reference | reference |
| | Episiotomy | 695 (19.7) | 59 (8.4) | 2.01 (1.45–2.76) | 1.88 (1.32–2.64) |
| Re-suturing | Spontaneous tear | 49 (1.4) | 7 (14.3) | 3.61 (1.46–7.72) | 3.28 (1.23–7.36) |
| | Episiotomy | 23 0.7) | 6 (26.1) | 7.65 (2.72–18.77) | 7.16 (2.52–17.82) |
| BMI (kg/m$^2$) | | | | | |
| <18.5 | Spontaneous tear | 54 (1.6) | 3 (5.6) | 1.21 (0.29–3.38) | 1.23 (0.29–3.46) |
| | Episiotomy | 31 (0.9) | 4 (12.9) | 3.04 (0.88–8.03) | 2.91 (0.84–7.72) |
| 18.5–24.9 | Spontaneous tear | 1 593 (47.0) | 74 (4.6) | reference | reference |
| | Episiotomy | 445 (13.1) | 48 (10.8) | 2.48 (1.69–3.62) | 2.27 (1.52–3.36) |
| ≥25.0 | Spontaneous tear | 1 058 (31.2) | 43 (4.1) | 0.87 (0.59–1.27) | 0.87 (0.59–1.27) |
| | Episiotomy | 210 (6.2) | 11 (5.2) | 1.13 (0.56–2.09) | 1.07 (0.53–1.98) |

[a]Percentage within each subgroup that reported being dissatisfied or very dissatisfied.

[b]Adjusted for age, BMI and mode of delivery for all exposure groups apart from Mode of delivery (adjusted for age and BMI) and BMI (adjusted for age and mode of delivery).

unbearable dyspareunia was also higher among women with an episiotomy compared with a spontaneous tear. At one year, postpartum infection, scar dehiscence, re-suturing and perineal pain at eight weeks were strong risk factors for being dissatisfied or very dissatisfied and to a somewhat lesser extent with strong or unbearable dyspareunia.

The main strengths of this cohort-study is its large size of more than 5 000 women and the robust data collected via three reliable national health and quality registers with high coverage. The large number of participants enabled subgroup analysis. The response rate on the questionnaire was high at both eight weeks and at one year (77.2% and 69.1%), lessening the risk of selection bias although it is still possible that women with symptoms choose to participate to a

higher extent than women without symptoms. Additionally, the data were collected to the registers prospectively, reducing the risk of recall bias.

Register studies are limited by incomplete coverage and also by missing values that could be of importance. Lack of data on pre-pregnancy levels of dyspareunia is a limitation. The PLR includes a pre-delivery questionnaire that includes questions on dyspareunia however the response rate is unfortunately too low to be useful. The fact that not every participating clinic reported all their second-degree tears and episiotomies, may have introduced selection bias as it is likely that more severe cases were included. This study lacks information on breastfeeding, however despite causing a hypoestrogenic state studies have not found an association between breastfeeding and dyspareunia [5, 22, 23]. In Sweden, approximately 30% of women partially breastfeed at 12 months [24]. In this study, there were no data available on the reasons why some respondents replied 'non-relevant' to the question regarding dyspareunia at one year. Unfortunately the response rate on whether continuous or interrupted sutures were used were low in this study, however a recent meta-analysis suggest that there is no difference in dyspareunia between the two methods [25]. There was also no specific data available on whether the satisfaction with the outcome of the perineal repair referred to function, cosmetic result or healing process. However, the majority of the questions in the questionnaire concern function.

In our study the prevalence of dyspareunia at one year following childbirth was 33.5% among women with spontaneous tear and 35.1% among those with episiotomy, which was slightly higher than the prevalence of 30% seen in the Australian Maternal Health Study. However they also included women with no tear and women with cesarean section [26]. In contrast, the Irish MAMMI cohort study of 832 women, of whom approximately half had a second-degree tear or an episiotomy, found a prevalence of dyspareunia at one year of 22% after second-degree tears and 24% after episiotomies, which was actually lower than their self-reported pre-pregnancy level of dyspareunia [27]. Other studies have reported a prevalence between 16% and 33%, however none of these studies looked specifically on second-degree tears [28–31]. The variation in prevalence in the literature may be due to a discrepancy in the extent of trauma as well as the definition used. We chose to include mild pain at penetrative intercourse, which may account for the higher prevalence of dyspareunia seen in our study.

Our finding, that women with episiotomy more often experience dyspareunia, is supported by a recent meta-analysis on 1210 women from five studies, that found increased the risk of dyspareunia by 65% [10]. The International Federation of Gynecology and Obstetrics recommends that episiotomy should be used in situations with a clear indication on the basis of increased risk for serious perineal tears, increased risk for perineal tear in a subsequent delivery and decrease in pelvic floor muscle strength [32]. In this study 38% of women with episiotomy compared to 14% of those with spontaneous tear, were delivered by vacuum suggesting that they may have had a more complicated second-stage. Labor and postpartum complications were also significantly more common in the episiotomy group. McDonald et al identified instrumental delivery as a risk factor for dyspareunia at 18 months [33]. In addition, there is evidence that vacuum-assisted delivery is associated with increased risk of wound infection and scar dehiscence, and that the routine use of antibiotics reduces the risk of both by approximately 50% [34]. Antibiotic prophylaxis in vacuum-assisted deliveries was not routinely given in Sweden at the time of this study. It is well established that vacuum-delivery is a risk factor for obstetric anal sphincter injury (OASI) and observational studies have shown that episiotomy may reduce that risk among nulliparous women delivered by ventouse by 25%-90% [35–38]. It is routine practice to consider episiotomy in ventouse delivery of nulliparas in Sweden and in 2021 the rate of episiotomies in vacuum-delivery was 33% (range 7–77%) [39].

The research available on other risk factors for dyspareunia identified in our study; perineal pain at eight weeks, postpartum infection, scar dehiscence and resuturing in relation to

dyspareunia is scarce or non-existent. The Australian Maternal Health Study found that women with perineal pain at one month postpartum had a 2.5 times higher odds of dyspareunia at six months [26]. Rosen et al showed in their systematic review that the predictors identified in previous studies have in many cases been equivocal and inconsistent [40]. Factors such as pre-pregnancy dyspareunia, mode of delivery, lack of vaginal lubrication, breast feeding and age have been identified in some studies while other studies have not been able to show an association [40].

To our knowledge, this is the first study that considers self-reported level of satisfaction of the outcome at one year after second-degree tear. It is known that OASI have a negative effect on quality-of-life and women's' self-reported outcome, however less is known after a second-degree tear [17, 41]. A Swedish qualitative study of women with second-degree tears found that a substantial number of participants experienced pain similar to those with OASI at eight weeks postpartum and that women ask for improved information as well as follow-up [42].

Our study also highlighted the problem with the disparity in perineal trauma among women with second-degree tears, from minor trauma to extensive injury, which may contribute to the variation in degrees of dyspareunia and level of satisfaction among the participants. In Sweden, a more detailed distinction of second-degree tears that include levator ani injuries was included in the ICD 10 coding system in 2020. A more comprehensive classification of second-degree tears will facilitate future research on the natural history, as well as predictors for poor outcomes among women with this heterogeneous group of injuries.

## Conclusion

For patients and clinicians, the results from this study shows that second-degree tears, including spontaneous tears and episiotomies, are associated with strong dyspareunia in only 2–4% of women at one year after delivery while around 70% of women report being satisfied or very satisfied with their outcome. While this knowledge is reassuring, some women will experience complicated second-degree tears despite efforts already in place to minimize the extent of the perineal trauma. Considering that postpartum infection, scar dehiscence, resuturing and perineal pain at eight weeks were identified as risk factors for strong or unbearable dyspareunia and dissatisfaction at one year, particularly in participants with episiotomy, it is suggested that women with several risk factors may benefit from postnatal follow-up such as a clinical appointment with a physician or a physiotherapist. More research is needed to evaluate the impact of postpartum interventions to minimize dyspareunia and to improve satisfaction.

## Acknowledgments

The authors would like to thank all participants in the Perineal Laceration Registry who took their time to complete the questionnaires.

## Author Contributions

**Conceptualization:** Maria Jonsson.

**Data curation:** Maria Jonsson.

**Formal analysis:** Mette L. Josefsson.

**Funding acquisition:** Mette L. Josefsson, Maria Jonsson.

**Methodology:** Mette L. Josefsson, Sara Sohlberg, Cecilia Ekéus, Eva Uustal, Maria Jonsson.

**Resources:** Maria Jonsson.

**Supervision:** Maria Jonsson.

**Writing – original draft:** Mette L. Josefsson.

**Writing – review & editing:** Mette L. Josefsson, Sara Sohlberg, Cecilia Ekéus, Eva Uustal, Maria Jonsson.

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
