## [Decision Letter · Decision Letter 0]

27 Aug 2024

PONE-D-24-25181Patient-reported dyspareunia and satisfaction with the outcome after spontaneous second-degree tear compared to episiotomy: a register-based cohort studyPLOS ONE Dear Dr. Josefsson,

Thank you for submitting your manuscript to PLOS ONE. After careful consideration, we feel that it has merit but does not fully meet PLOS ONE’s publication criteria as it currently stands. Therefore, we invite you to submit a revised version of the manuscript that addresses the points raised during the review process.

We look forward to receiving your revised manuscript.

Kind regards,

Fereshteh Behmanesh, PhD

Academic Editor

PLOS ONE

“The authors would like to thank all participants in the Perineal Laceration Registry who took their time to complete the questionnaires. This research was funded by the Region Uppsala Research and Development Grant and by the General Maternity Hospital Foundation.”

“MLJ received funding by the Region Uppsala Research and Development Grant and by the General Maternity Hospital Foundation. The sponsors did not play any role in the study design, data collection and analysis, decision to publish, and preparation of the manuscript.”

3. Please note that your Data Availability Statement is currently missing the repository name and/or the DOI/accession number of each dataset OR a direct link to access each database. If your manuscript is accepted for publication, you will be asked to provide these details on a very short timeline. We therefore suggest that you provide this information now, though we will not hold up the peer review process if you are unable.

Additional Editor Comments:

Dear Author

The reviewer’s comments are below:

These are my comments for this paper:

1. In the summary, the year and the place of the study should be mentioned.

2. In the conclusion, a summary should be written “considering that..............................., it is suggested …. .... .... (write your suggestions for clinical practice and scientific research )

3. In Table 3, bold the main variables to distinguish them from the sub-variables, such as : delivery mood, pain at 8 weeks, postpartum infection, etc., or use indentation.

4. Although breastfeeding causes vaginal dryness, it does not lead to reportable dyspareunia, however, please find more references for this issue, otherwise delete the sentence.

5. According to the very good study that has been done and the insight that the researchers have found on the subject, it is suggested to propose a questionnaire development study for designing a tool investigated the consequences of spontaneous and non-spontaneous vaginal tears so that it can be used by other researchers and also policymakers could be familiar with further variables that must be included into the registries.

Comments to the Author

-Unfortunately, in my opinion, this article hasn't any novelty and is one of the most repeated articles that have been studied from all aspects.

- There is no innovation, the results are predictable before the evaluation A study with a larger sample size showed that episiotomy causes dyspareunia. What is the gap of the study? The results are predictable before the review

- According to Table 1, two groups have significant differences in terms of labor and postpartum complications, which affects the results

-- The conclusion should be written as a statement, not repeating the results

- Grammar errors and languages. There are many grammar errors in the manuscript that require substantial improvement.

Reviewers' comments:

Reviewer's Responses to Questions

**Comments to the Author**

1. Is the manuscript technically sound, and do the data support the conclusions?

Reviewer #1: Yes

Reviewer #2: Partly

2. Has the statistical analysis been performed appropriately and rigorously? 

Reviewer #1: Yes

Reviewer #2: Yes

3. Have the authors made all data underlying the findings in their manuscript fully available?

Reviewer #1: Yes

Reviewer #2: Yes

4. Is the manuscript presented in an intelligible fashion and written in standard English?

Reviewer #1: Yes

Reviewer #2: Yes

5. Review Comments to the Author

Reviewer #1: These are my comments for this paper:

1. In the summary, the year and the place of the study should be mentioned.

2. In the conclusion, a summary should be written “considering that..............................., it is suggested …. .... .... (write your suggestions for clinical practice and scientific research )

3. In Table 3, bold the main variables to distinguish them from the sub-variables, such as : delivery mood, pain at 8 weeks, postpartum infection, etc., or use indentation.

4. Although breastfeeding causes vaginal dryness, it does not lead to reportable dyspareunia, however, please find more references for this issue, otherwise delete the sentence.

5. According to the very good study that has been done and the insight that the researchers have found on the subject, it is suggested to propose a questionnaire development study for designing a tool investigated the consequences of spontaneous and non-spontaneous vaginal tears so that it can be used by other researchers and also policymakers could be familiar with further variables that must be included into the registries.

Reviewer #2: Comments to the Author

-Unfortunately, in my opinion, this article hasn't any novelty and is one of the most repeated articles that have been studied from all aspects.

- There is no innovation, the results are predictable before the evaluation A study with a larger sample size showed that episiotomy causes dyspareunia. What is the gap of the study? The results are predictable before the review

- According to Table 1, two groups have significant differences in terms of labor and postpartum complications, which affects the results

-- The conclusion should be written as a statement, not repeating the results

- Grammar errors and languages. There are many grammar errors in the manuscript that require substantial improvement.

6. PLOS authors have the option to publish the peer review history of their article (what does this mean?). If published, this will include your full peer review and any attached files.

Reviewer #1: **Yes: **ok

Reviewer #2: No

---

## [Author Response · Author response to Decision Letter 0]

10 Oct 2024

Dear Editors, 

thank you very much for your time and effort with our manuscript. Your questions and comments have been helpful for us to make improvements and alterations to our manuscript. Our answers to the comments and questions from the reviewers have been addressed in detail in the document titled 'Response to reviewers'.

---

## [Decision Letter · Decision Letter 1]

3 Dec 2024

Patient-reported dyspareunia and outcome satisfaction after spontaneous second-degree tear compared to episiotomy: a register-based cohort study

PONE-D-24-25181R1

Dear Dr. Mette L Josefsson

We’re pleased to inform you that your manuscript has been judged scientifically suitable for publication and will be formally accepted for publication once it meets all outstanding technical requirements.

Kind regards,

Fereshteh Behmanesh, PhD

Academic Editor

PLOS ONE

Additional Editor Comments (optional):

Reviewers' comments:

Reviewer's Responses to Questions

**Comments to the Author**

1. If the authors have adequately addressed your comments raised in a previous round of review and you feel that this manuscript is now acceptable for publication, you may indicate that here to bypass the “Comments to the Author” section, enter your conflict of interest statement in the “Confidential to Editor” section, and submit your "Accept" recommendation.

Reviewer #3: All comments have been addressed

2. Is the manuscript technically sound, and do the data support the conclusions?

Reviewer #3: Yes

3. Has the statistical analysis been performed appropriately and rigorously? 

Reviewer #3: Yes

4. Have the authors made all data underlying the findings in their manuscript fully available?

Reviewer #3: Yes

5. Is the manuscript presented in an intelligible fashion and written in standard English?

Reviewer #3: Yes

6. Review Comments to the Author

Reviewer #3: Dear editor of PLOS ONE

This manuscript id “Patient-reported dyspareunia and outcome satisfaction after spontaneous second degree tear compared to episiotomy: a register-based cohort”

As the authors state, sexual function in women had a key role in women’s quality of life and dyspareunia is a problem that affects women's lives. Studies in this field are valuable.

- In the Method and material section, On line 103, enter a reference for the BMI classification.

7. PLOS authors have the option to publish the peer review history of their article (what does this mean?). If published, this will include your full peer review and any attached files.

Reviewer #3: No

---

## [Editor Report · Acceptance letter]

9 Dec 2024

PONE-D-24-25181R1 

PLOS ONE

Dear Dr. Josefsson, 

I'm pleased to inform you that your manuscript has been deemed suitable for publication in PLOS ONE. Congratulations! Your manuscript is now being handed over to our production team.

Kind regards, 

on behalf of

Dr. Fereshteh Behmanesh 

Academic Editor

PLOS ONE